# Scalable and Quench-Free Processing of Metal Halide Perovskites in Ambient Conditions

Carsen Cartledge [1], Saivineeth Penukula [2], Antonella Giuri [3], Kayshavi Bakshi [1], Muneeza Ahmad [1], Mason Mahaffey [2], Muzhi Li [1], Rui Zhang [1], Aurora Rizzo [3] and Nicholas Rolston [2,*]

1   School for Engineering of Matter, Transport and Energy, Ira A. Fulton Schools of Engineering, Arizona State University, Tempe, AZ 85281, USA; cacartle@asu.edu (C.C.); kbakshi@asu.edu (K.B.); mahmad15@asu.edu (M.A.); muzhili1@asu.edu (M.L.); rui.zhang.10@asu.edu (R.Z.)
2   School of Electrical, Computer and Energy Engineering, Ira A. Fulton Schools of Engineering, Arizona State University, Tempe, AZ 85281, USA; spenukul@asu.edu (S.P.); mpmahaff@asu.edu (M.M.)
3   CNR NANOTEC—Institute of Nanotechnology, c/o Campus Ecotekne, University of Salento, Via Monteroni, I-73100 Lecce, Italy; antonella.giuri@nanotec.cnr.it (A.G.); aurora.rizzo@nanotec.cnr.it (A.R.)
*   Correspondence: nicholas.rolston@asu.edu

**Abstract:** With the rise of global warming and the growing energy crisis, scientists have pivoted from typical resources to look for new materials and technologies. Perovskite materials hold the potential for making high-efficiency, low-cost solar cells through solution processing of Earth-abundant materials; however, scalability, stability, and durability remain key challenges. In order to transition from small-scale processing in inert environments to higher throughput processing in ambient conditions, the fundamentals of perovskite crystallization must be understood. Classical nucleation theory, the LaMer relation, and nonclassical crystallization considerations are discussed to provide a mechanism by which a gellan gum (GG) additive—a nontoxic polymeric saccharide—has enabled researchers to produce quality halide perovskite thin-film blade coated in ambient conditions without a quench step. Furthermore, we report on the improved stability and durability properties inherent to these films, which feature improved morphologies and optoelectronic properties compared to films spin-coated in a glovebox with antisolvent. We tune the amount of GG in the perovskite precursor and study the interplay between GG concentration and processability, morphological control, and increased stability under humidity, heat, and mechanical testing. The simplicity of this approach and insensitivity to environmental conditions enable a wide process window for the production of low-defect, mechanically robust, and operationally stable perovskites with fracture energies among the highest obtained for perovskites.

**Keywords:** perovskite solar cells; blade coating; polymer-mediated crystallization; nucleation; GIWAXS; passivation; lock-in thermography; mechanical stability; defect tolerance; gellan gum

## 1. Introduction

The halide perovskite (PVSK) family of materials proves to be the rising star in the next generation of solar devices, with lab-reported power conversion efficiencies (PCE) rivaling commercial silicon modules at 26.1% after just over ten years of research [1]. Additionally, the compositionally tunable bandgap of PVSK films offers an attractive means of advancing silicon modules when used in tandem, making the push towards scalability increasingly urgent. While PVSK technologies show great promise, the U.S. Department of Energy continues to consider manufacturability, stability, and scalability as key challenges in the effort to reach commercialization [2]. Furthermore, large-area perovskite modules prove even more challenging in their development, with efficiency gaps appearing between them and their small-scale counterparts under 200 cm$^2$ [3].

In an effort to address these challenges and move from lab-to-fab, emphasis has shifted from processing via spin coating to innovating scalable techniques such as spray coating,

electrochemical deposition, ink jet printing, slot-die coating, and doctor-blade coating [4–6]. Specifically, meniscus-driven doctor-blade coating offers a promising lab-scale deposition method that easily translates to industrial manufacturing techniques such as slot-die coating and roll-to-roll manufacturing on flexible substrates. Interestingly, blade coating and slot-die coating are suggested to be among the most cost-affordable and promising solution processing technologies for industrialization [7].

However, this transition from spin coating to meniscus-driven processing is often complicated by an incompatibility with inks optimized for use with traditional antisolvent quenching. Tuning parameters associated with alternative quenching methods, such as gas or vacuum-based techniques, further complicate process optimization, reproducibility, and the overall feasibility of commercialization. Entire studies have been dedicated to the reproducibility of these methods recently [8].

Furthermore, transitioning from lab-style inert fabrication to industrial ambient manufacturing poses further complications. Open-air, antisolvent-free blade coating of unmodified PVSK inks previously optimized for spin coating in an inert environment with an antisolvent produces void-filled, uneven, and shunted films, demonstrating that the mild annealing step alone is not enough to control crystallization [4,9,10]. In recent studies, solvent engineering and novel polymeric modifiers have been used to improve film uniformity, density, and stability while also improving manufacturability through an increase in ink viscosity, which offers improved fluid dynamics and crystallization while blade coating [11–13]. Furthermore, biopolymers from the food industry, such as cornstarch, have been found to enhance mechanical integrity and the operational lifetime of devices in addition to inducing spherulitic domains that can be tuned in size by temperature and precursor concentration to increase PCEs, while a hypothesized organizational scaffolding increases resistance to humidity [14–16].

The biopolymer chosen for this study, gellan gum—a non-toxic extracellular polysaccharide produced from bacteria known as *Sphingomonas elodea*—forms firm, transparent gels in the presence of metallic ions and features heat resistance properties [17,18]. Gellan gum is thought to mediate the gelation process by undergoing a reversible transition from a disordered single chain to a self-assembled highly ordered double helix upon cooling through crosslinking structures, whereby various fibrous formations are possible depending on the temperature, cation attachment, and concentration introduced [18]. Outside of being used as a suspension agent, binder, and coagulant in the food industry, gellan gum has been utilized in the biomedical community and conductive polymer field to create high-quality films without interrupting expected electrical properties [19,20]. More recently, gellan gum has been incorporated into solar devices featuring a polymer-mediated crystallization process to achieve single-junction methylammonium lead iodide, $CH_3NH_3PbI_3$ (MAPI), devices, and wide-bandgap mixed-halide perovskite solar cells for tandem applications with improved photostability compared to reference cells. For MAPI devices with gellan gum, Bisconi et al. reported an efficiency of 16.98% with an open-circuit voltage of 1.072, short circuit current density of 21.24 mA/cm$^2$ and fill factor of 74.6%. For $MAPb(I_{0.8}Br_{0.2})_3$ devices manufactured with gellan gum, they reported an efficiency of 16.37% with an open-circuit voltage of 1.104, short-circuit current density of 20.36 mA/cm$^2$, and fill factor of 72.8% [21]. The addition of gellan gum has also been shown to induce an intrinsic compressive stress within the perovskite film, a property correlated with improved mechanical and environmental stability [22,23].

Rather than directly controlling solvent evaporation with an antisolvent, air gun, or the use of heat, gellan gum alters fluid dynamics and free energy considerations within the ink to tune the crystallization process by acting as a barrier to excessive homogenous and early heterogeneous nucleation through intermediate interactions and a change in wetting angle thanks to rheological modification. These factors stimulate the radial, spherulitic growth of similarly oriented domains that grow in a space-filling nature that combats the presence of pinholes while possibly collecting defects and impurities, including the additive, along grain boundaries, which may assist in relieving detrimental thermal or

intrinsic stresses typically associated with the thermal expansion mismatch between the PVSK layer and substrate without inhibiting optoelectronic properties [12,22].

In this work, we demonstrate a method of polymer-mediated crystallization enabling the production of high-quality blade-coated MAPI films in open air without quenching that feature improved mechanical, optoelectronic, and stability characteristics. With the introduction of less than one weight percent of gellan gum, we show that the crystallization process can be fully controlled, which allows for the tuning of the perovskite film morphology according to the additive concentration. This tunability is enabled by controlling supersaturation rates of the evaporating wet film, which thereby changes the balance between the nucleation and growth regimes involved in the overall crystallization process [10,20].

## 2. Materials and Methods

All of the materials were purchased and used as received unless otherwise stated. Methylammonium Iodide (MAI, >99.99%), $CH_3NH_3I$, was purchased from GreatCell Solar (Queanbeyan, Australia). Lead (II) Iodide, $PbI_2$ (99.99%), was purchased from TCI (Tokyo, Japan). Gellan gum was sourced from Alfa Aesar (Ward Hill, MA, USA) and was oven-dried at 80 °C for 3 days before use and then stored in a nitrogen dry box. The solvent, dimethyl sulfoxide (DMSO, ≥99%), was purchased from Sigma Aldrich (Burlington, MA, USA). Gellan gum powder was purchased from Alfa Aesar.

The halide PVSK ($CH_3NH_3PbI_3$) precursor solution was prepared by mixing MAI and $PbI_2$ in a molar ratio of 1:1 in 1 mL of DMSO in a nitrogen glovebox. This solution was then mixed with a stir bar at 80 °C for 30 min. This stock solution was pipetted into separate vials for further DMSO dilution to 0.5 M and the addition of various amounts of dried gellan gum, such as 0.18% and 0.56% weight percent of the perovskite precursors. The gellan gum MAPI inks were heated for 2 more hours at 80 °C while being mixed continuously with a stir bar until complete visual dissolution of the gellan gum particles occurred.

Plain or patterned ITO glass substrates were washed with detergent and rinsed with deionized water before being sonicated sequentially in acetone and isopropanol for 5 min each. The substrates were dried with a nitrogen gun and UV-ozone treated for 10 min before coating. When investigating gum concentrations of 0.56% or less, a ten-minute substrate pre-heating period was allowed if coating on a heated substrate that did not exceed 55 °C. The blade coater gap varied from 100–300 μm to accommodate for variations in the glass substrate and ink viscosity, while blade speeds ranged from 2.5 mm/s to 30 mm/s. Spin coating was done in a nitrogen glovebox at 4000 rpm for 30 s with chlorobenzene as the antisolvent. All films were annealed at 100 °C for 25 min. A breakdown of the characterization techniques utilized is provided in Appendix A.

## 3. Results and Discussion

We tested several concentrations of MAPI-x% gellan gum (GG), where x% GG is the weight percent of gum to perovskite precursors. For example, 0.18% corresponds to 1.1 mg of GG in 1 mL of MAPI. Note that the ratio of perovskite precursor to dimethyl sulfoxide (DMSO) remained the same for all inks, 28 wt%. As seen in Figure 1, higher concentrations of GG, such as 2.29%, approach the viscosity of honey and feature stronger pseudo-plastic behavior, which requires significantly slower coating speeds (below 2.5 mm/s) to achieve the desired 500-nm-thick films compared to lower gum concentrations like 0.56% and 0.18%, which behave more like Newtonian fluids with a constant viscosity at a given temperature and can be coated at faster speeds (upwards of 25 mm/s) that are needed for commercial viability. In addition to altering processing parameters, an increased concentration of GG slows film conversion by altering the crystallization process and the resulting final film morphology due to the polymer's interactions with the solvent and precursor materials. Additionally, GG acts as an especially efficient thickener compared to other polysaccharide additives such as cornstarch, which require more than 10 wt% to achieve similar shear viscosities to 0.56% GG [14].

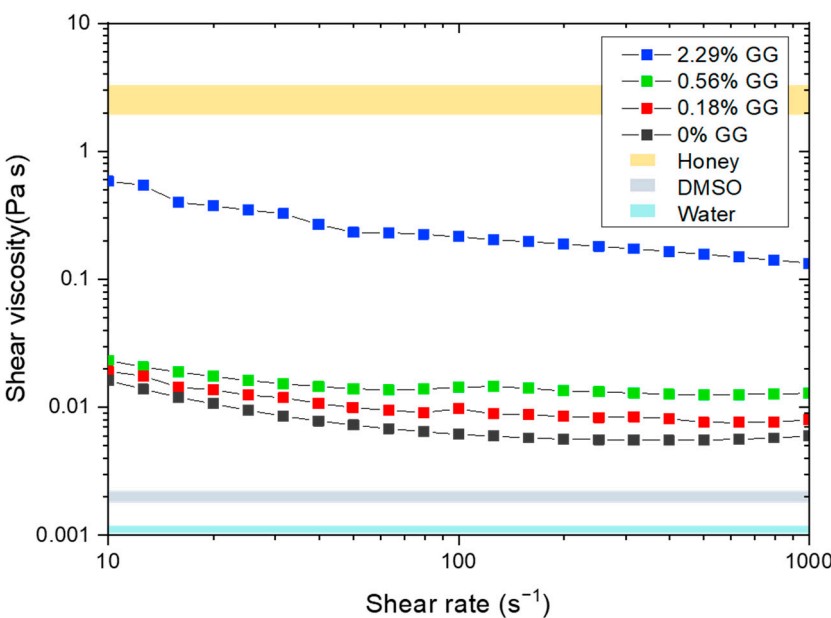

**Figure 1.** Rheological data demonstrating the effects of different concentrations of GG to MAPI ink with respect to honey, water, and the pure DMSO.

After identifying the rheological properties of GG, we then studied the effect of viscosity on PVSK crystallization. While the exact mechanisms of the PVSK crystallization process remain an active area of research, the LaMer theory, a qualitative analogy that builds upon classical nucleation theory to describe the removal of solvent from a simplified monodispersed crystalline nanoparticle system, is often presented as a link between theory and processing [6,16,24–29]. The LaMer curve is most often plotted with concentration versus time respective to the nucleation and growth regimes to illustrate their respective dominance and influence on morphology, yet this plot is not always intuitive from a processing point of view. Figure 2a presents a modified visualization of the classic LaMer curve, where concentration is plotted against processing time for complete film conversion. The plot begins with the initial drying process, whereby the removal of solvent from the wet film increases the concentration of the remaining liquid to saturation, $C_s$, at a near-constant rate before reaching a critical supersaturation concentration, $C^*$, between $C^*_{min}$ and $C^*_{max}$, whereby stable nucleates are likely to form. Once nucleation occurs, the available precursor concentration in solution will drop below $C^*_{min}$ into a regime dominated by crystal growth until all available liquid is removed. Because the final morphology of the PVSK is dependent on the balance of the nucleation and growth regimes, controlling the rate of supersaturation during processing is of utmost importance.

For example, the red curve in Figure 2a corresponds to spin coating with an antisolvent, which induces a sudden film conversion due to the high rate and degree of supersaturation achieved, which results in instantaneous burst nucleation because of the targeted solvent removal. This fast, uniform, and dense burst nucleation scenario creates fine-grained morphologies such as the film in Figure 2b,g, which were captured via tapping mode atomic force microscopy (AFM) and transmission optical microscopy, respectively. On the other extreme, as seen in grey in Figure 2a, if the degree and rate of supersaturation are low, then the conversion of the film will be slow due to a lesser driving force in the creation of stable nucleates as prolonged growth will occur at these fewer existing sites. This slow process yields rough dendritic films with many voids, like the antisolvent-free blade-coated 0% GG film in Figure 2c,h.

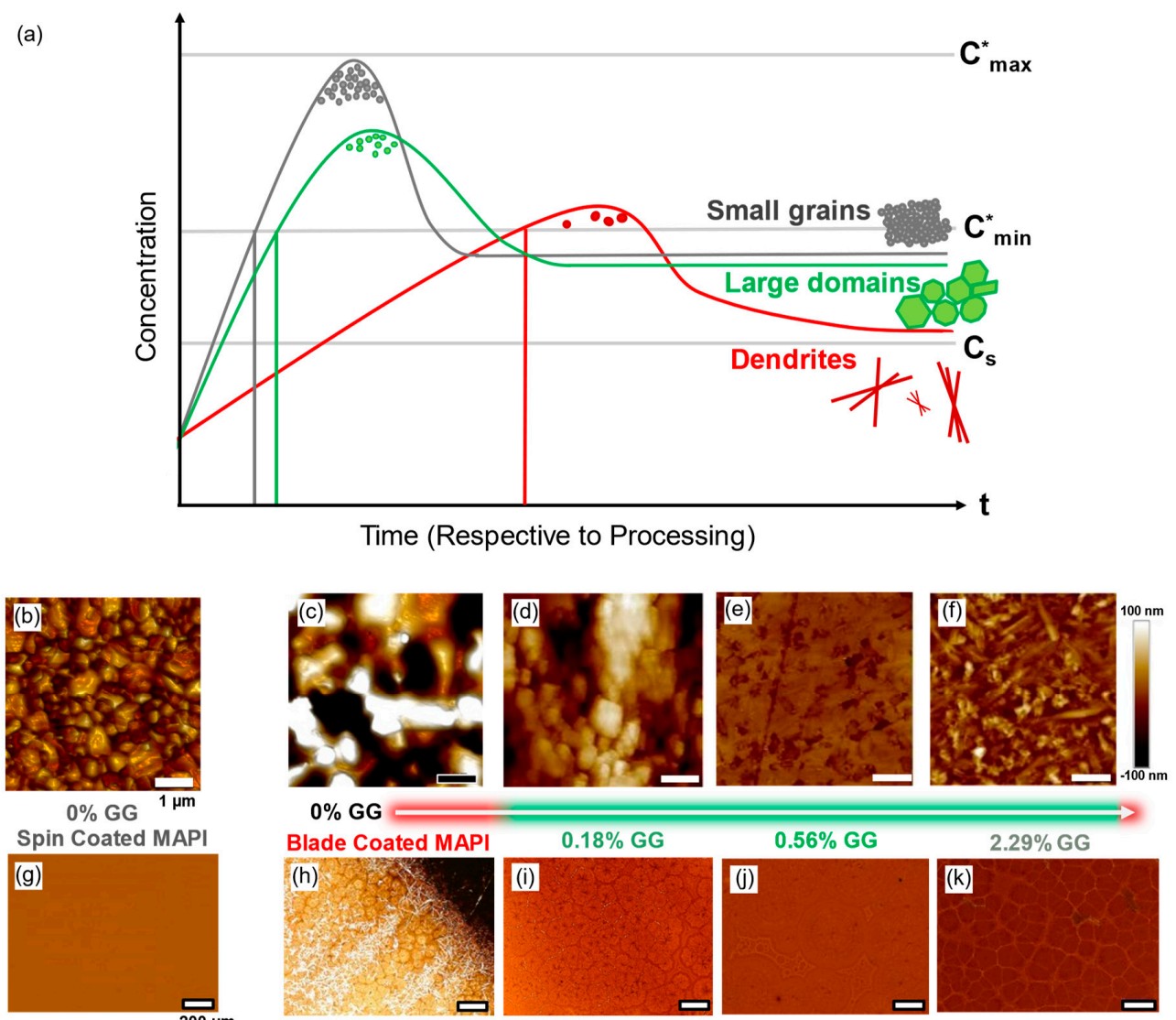

**Figure 2.** (**a**) Modified LaMer curve depicting supersaturation rates as a function of precursor concentration in wet films undergoing drying respective to processing time. The red curve represents a case of rapid supersaturation with burst nucleation and little time for crystal growth, which is characteristic of antisolvent-induced crystallization, the green line represents the introduction of an optimized amount of GG, whereby burst nucleation occurs at a lower degree of supersaturation, thus allowing more time for crystal growth which yields larger domains, and the grey line represents a case where fewer stable nucleates form, leading to extended crystal growth on available sites resulting in a void-filled dendritic structure; (**b**) AFM image of fine-grained MAPI spin-coated with an antisolvent in a nitrogen glovebox; (**c**–**f**) AFM images of ambient blade-coated samples with increasing amounts of GG from unmodified to 2.29%, showing the change in morphology from variable dendrites to homogeneous domains as suggested by the LaMer curve; (**g**–**k**) Corresponding transmission optical microscopy images of (**b**–**f**) further demonstrating the uniformity and changes in morphology enabled by GG. All transmission images were captured with the same lighting conditions.

GG enables modulation between these two extremes. The green curve in Figure 2a corresponds to the polymer-mediated process where sufficiently dense burst nucleation ensures complete coverage while reserving time and nutrients for domain growth. The AFM image in Figure 2d and the corresponding transmission image in Figure 2i show the increase in size of similarly oriented domains with 0.18% GG versus ultra-fast antisolvent quenching in Figure 2b, as suggested by the LaMer curve. Likewise, at an optimized concentration

of 0.56% GG, AFM images in Figure 2e and the corresponding transmission image in Figure 2j reveal that large, uniform spherulitic domains and a compact surface morphology are favored, which may create a better interface with transport layers deposited after the active layer in solar devices. Interestingly, AFM and transmission images in Figure 2f,k demonstrate that a high concentration of GG can delay the onset of both homogeneous and heterogeneous nucleation much longer than 0.56% GG ink, with one such source of this delay resulting from the increased viscosity, which can slow species diffusion rates critical to the overall crystallization process, such that even a lower supersaturation rate stimulates a dense burst-like nucleation event accompanied by extra time for crystallite growth into textured dendritic microstructures while retaining the overall morphology and uniformity of spherulitic domains. While this surface texturing is accompanied by visually continuous films, 0.56% GG was selected as the upper threshold for our following experiments due to its decrease in root-mean-square roughness from 19.2 nm in the spin-coated control film to 11.2 nm, which offers a promising interface for the deposition of thin-charge transport layers that will be fabricated on top of the PVSK film. In summary, utilizing the lowest concentration of GG possible while maintaining sufficient morphological control over roughness, crystallinity, and uniformity during processing led to the selection of 0.56% GG and below for further investigation thanks to the increased processing speeds inherent to the fluid dynamics of the mildly viscous inks compared to higher concentrations such as 2.29%, as well as to avoid unintended electronic consequences as the insulator-like tendency of the polymeric GG could disrupt efficient charge flow above a critical concentration.

We then performed optoelectronic characterization of the MAPI films as a function of GG concentration. Illuminated lock-in thermography (ILIT) was thus performed on blade-coated samples with 0%, 0.18%, and 0.56% GG to observe the correlation between morphology changes in the film and thermographic response with the compositions. Thermal topography images from Figure 3a–c validate the improvement in the uniformity of the deposited film with the addition of the optimum amount of the additive (0.56% GG) to the ink, which is observed as a lesser variation of temperature across the sample as shown in Figure 3c. Variations in temperature or the presence of dark and bright regions in Figure 3a,b imply the presence of gaps or morphological variations across the samples observed in the samples with 0% GG and 0.18% GG. Illumination amplitude images from Figure S1a–c in correlation with illumination phase images from Figure S1d–f validate the presence of these gaps and the improvement of film uniformity. The presence of extremely bright spots on the illuminated amplitude images corresponds to either recombination centers or defects. The phase variations on the illuminated phase images show the changes in the depth at those particular points. Based on the above correlations, the reduction in the bright spots on the amplitude image and the reduction in the phase changes on the phase image for the 0.56% GG composition show the improvement in uniformity and optoelectronic response across the sample. These improvements were further validated by electronic characterization when I-V response, series, and shunt resistances of these samples were measured (ITO/MAPI/Carbon), where 0% and 0.18% GG did not show any response to the applied voltage and 0.56% GG showed uniform I-V response (Figure 3d). The 0.56% GG condition also showed an improvement with the values of resistance being in the required range for a good IV response, with very low series resistance and very high shunt resistance (Table S2).

Next, we tested the effects of GG inclusion on the operational stability of MAPI films. X-ray diffraction spectrums and photoluminescent data were collected for unencapsulated MAPI blade-coated samples after humidity (25C, 85% RH in an environmental test chamber) or heat aging (85C in a vacuum oven as to avoid humidity-based degradation) initially and after three hours and 12 h, as shown in Figures 4 and 5, in which grey correlates to as-coated films before testing, blue to samples after 3 h of testing, and red for samples after 12 h of cumulative testing. Figure 4a–c indicates the effects of aging in humid air for 12 cumulative hours with both 0% GG (Figure 4a) and 0.18% GG (Figure 4b) featuring a near-complete elimination of characteristic perovskite peaks coupled with an increase in

peaks associated with lead iodide. This crystallographic evidence suggests the volatilization of the methylammonium cation, possibly as a means of stress relaxation, and PL data in Figure 4d,e confirms the near-complete loss of the black photoreactive phase after 12 h, as seen in the optical images alongside Figure 4a,b. In contrast, at an optimized concentration of the polymeric additive, 0.56% GG, the film retains its characteristic perovskite peaks and black phase (Figure 4c) while suppressing excessive lead iodide formation with notably improved retention of the photoreactive phase after 12 h of moisture exposure (Figure 4f). This improvement against humidity can likely be attributed to the improved film uniformity and morphology made possible by the alterations in film crystallization induced by the addition of the GG, and this evidence further points towards an induced beneficial compressive stress inherent to the film.

XRD and PL data were then collected for ambient MAPI blade-coated samples after aging under vacuum at 85 °C initially and after three hours and 12 h, as shown in Figure 5. While the XRD evidence of the degradation of the perovskite into lead iodide is not as dramatic as under humid conditions in Figure 4, which is plotted across a wide 2θ axis, Figure 5c indicates that 0.56% GG can markedly suppress lead iodide formation compared to 0% GG and 0.18% GG (Figure 5a,b). It is important to also note the dramatic difference in final film quality between 0% GG and the GG samples as seen in inset transmission camera images in Figure 5a–c, clearly indicating the improved coverage enabled by GG. The PL data (Figure 5d–f) enables a more nuanced evaluation of the optoelectronic degradation across the samples with aging, with both 0% and 0.18% GG films featuring a continuous drop in PL intensity while 0.56% GG retains a higher PL reading with no noticeable loss in intensity between 3 and 12 h of aging.

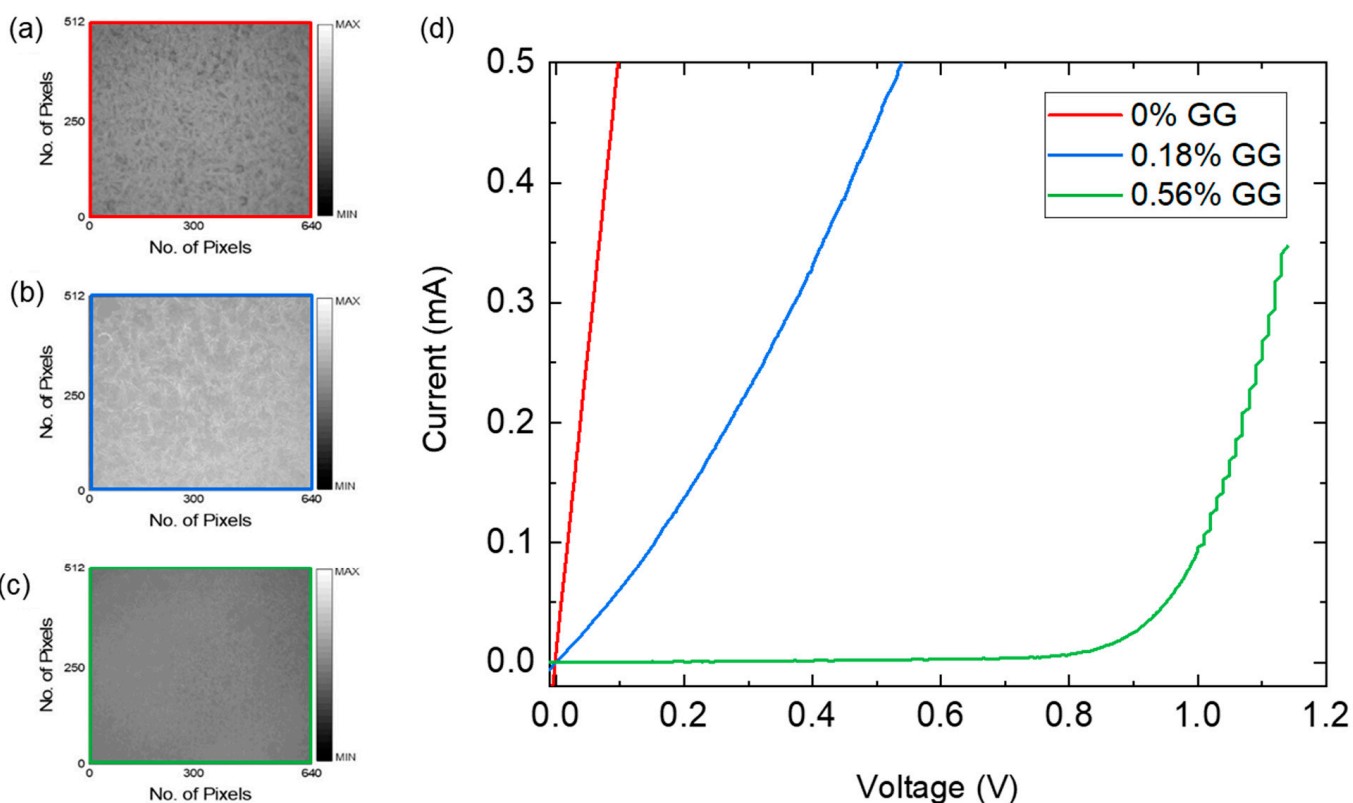

**Figure 3.** (**a–c**) Illuminated lock-in thermal topography images of 0%, 0.18%, and 0.56% GG with scale bar indicating minimum temperature as dark spot and maximum temperature as bright spot, all the images have a common scale of 4.6 μm per pixel; (**d**) I–V response of the blade coated thin films on ITO glass with 0% and 0.18% GG samples showing poor response versus 0.56% GG.

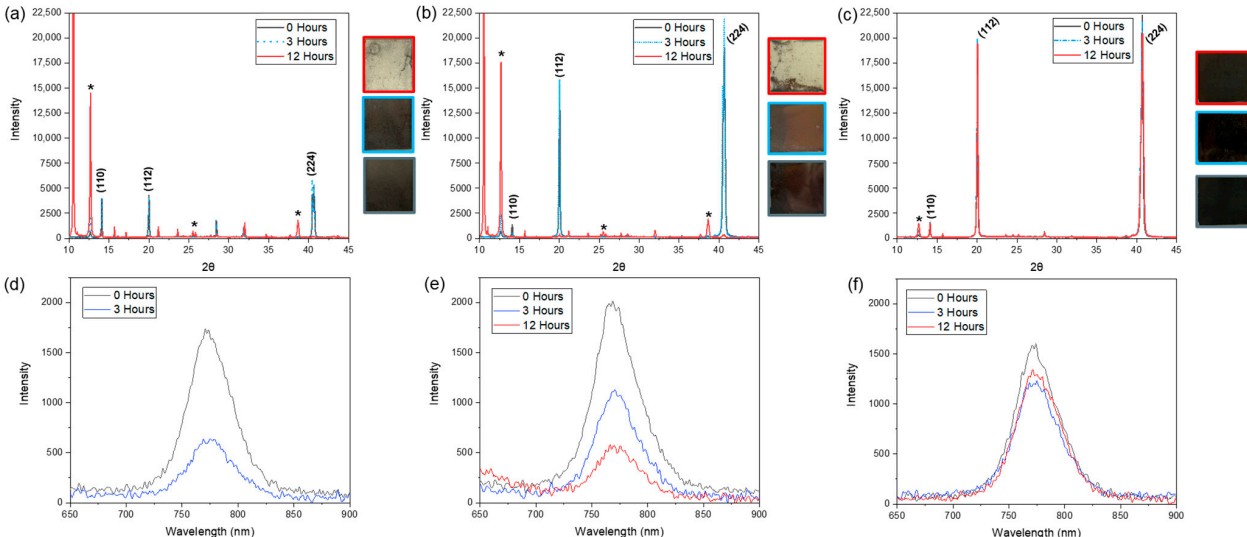

**Figure 4.** (**a**–**c**) XRD spectrums of ambient blade-coated MAPI with 0%, 0.18%, and 0.56% GG after 0, 3, and 12 total hours of humidity aging at 85% RH with accompanying color-coded macroscale camera photographs after each step in the aging process. Both the MAPI and 0.18% GG films near-complete loss of the black photoreactive phase both visually and as evidenced by the loss of characteristic perovskite peaks accompanied by a dramatic increase in $PbI_2$ peak intensities after 12 h while 0.56% GG retains its perovskite peaks with only slight $PbI_2$ peak intensification. Each XRD spectrum includes labeled dominant MAPI and $PbI_2$ peaks denoted by miller indices or a star, respectively; (**d**–**f**) corresponding PL spectrums confirming near-complete degradation of optoelectronic properties for MAPI and 0.18% GG films, while 0.56% GG retains a significant portion of its PL, indicating enhanced stability.

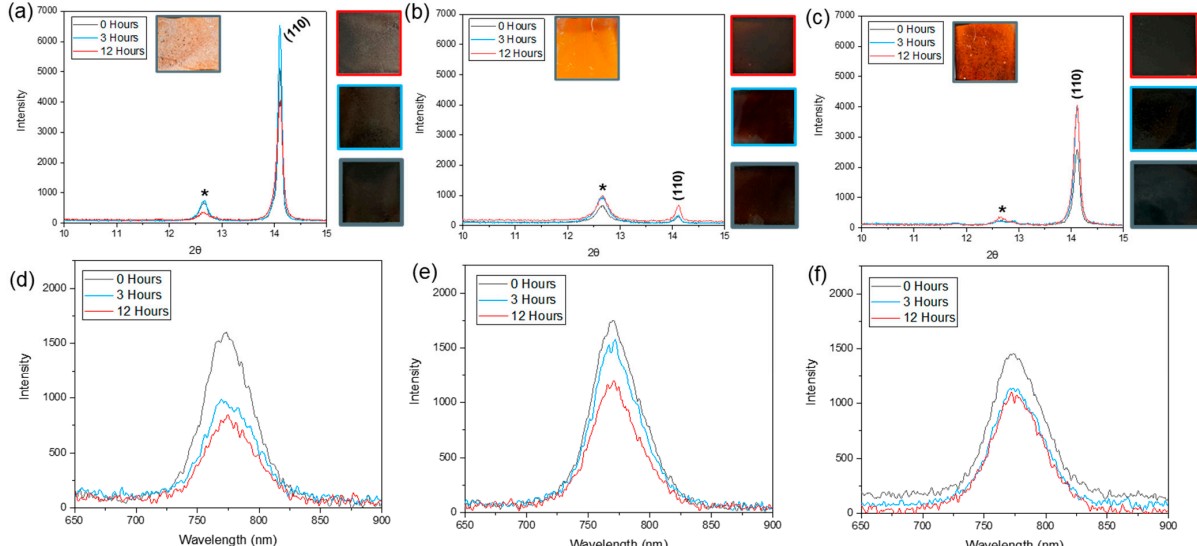

**Figure 5.** (**a**–**c**) XRD spectrums of ambient blade-coated MAPI with 0%, 0.18%, and 0.56% GG after 0, 3, and 12 total hours of heat exposure at 85 °C with accompanying color-coded macroscale camera photographs after each step in the aging process. Both the MAPI and 0.18% GG films feature a stronger increase in $PbI_2$ peak intensities after 12 h than 0.56% GG, and the improved film quality is further supported through the inset transmission photographs. Each XRD spectrum includes labeled dominant MAPI and $PbI_2$ peaks denoted by miller indices or a star, respectively, and Figure S2 provides these spectrums plotted on a wider 2θ axis; (**d**–**f**) Corresponding PL spectrums confirming the ability of 0.56% GG to slow degradation of optoelectronic properties compared to 0% and 0.18% GG films, indicating enhanced stability.

As seen in Figures 4a–c and S2, an increasing dominance of the (112) plane located at 2θ of approximately 20° as well as the corresponding (224) plane at approximately 40° over the traditionally dominant (110) MAPI plane in spin-coated samples at approximately 14° indicates that blade-coated GG films have a preferential orientation of the polycrystalline grains that likely arises at least in part to the uniaxial direction of coating. This change in orientation of the PVSK grains is consistent between the surface and bulk as supported by grazing incident wide-angle X-ray scattering (GIWAXS) data collected at various penetration depths, and the data confirms that an increase in GG leads to partially to highly oriented 3D bulk crystal texturing (Figure S3). This change in crystal orientation likely plays a role in degradation that is thought to correspond to stress relaxation in the perovskite film, and we hypothesize that defects present and diffusion along grain boundaries likely play a critical role in film degradation and its rate of occurrence. However, some defects may have beneficial effects, with previous evidence of the presence of self-passivated lead iodide between adjacent spherulites at the grain boundaries in blade-coated samples being reported, and this might explain the improved stability of GG films over 0% GG samples due to the intensity of lead iodide peaks present relative to PVSK peaks in the GG films at 0 h in Figures 4b,c and 5b,c [12].

In addition to the improved resistance to aging, we hypothesized that the mechanical properties of the GG films would be improved as well. Our previous work validated this in spin-coated PVSK with starch additives, where a significant increase in fracture energy was observed with the addition of 10 wt% of starch and above [15]. Double cantilever beam testing was performed on spin-coated control and blade-coated samples with varying compositions of GG to observe the changes in the fracture energy, $G_c$, of the MAPI films. Figure 6a indicates that the addition of GG can increase the average $G_c$ of the perovskite layer from $1.5 \pm 0.2 \, J/m^2$ for a spin-coated control to $6.6 \pm 2.5 \, J/m^2$ with the addition of 0.18% GG and $10.9 \pm 2.3 \, J/m^2$ for 0.56% GG, which is among the highest values obtained for perovskites and is comparable with crystalline silicon modules that have a $G_c$ of ~10 $J/m^2$ [30]. A $G_c$ of over 5 $J/m^2$ has been identified as a critical metric for ensuring durability during handling, a value which is exceeded by both 0.18% and 0.56% GG films [15,30]. Note that a spin-coated 0% GG sample was chosen over a blade-coated one due to the many voids present in the latter (see Figure 2c,h), which results in fracture outside of the PVSK yielding unrepresentative fracture energies similar to that of the epoxy layer. Mitigating delamination and fracture of the active layer is critical to increasing the overall durability and stability of PVSK solar cells, especially with respect to suppressing pathways to accelerated environmental degradation.

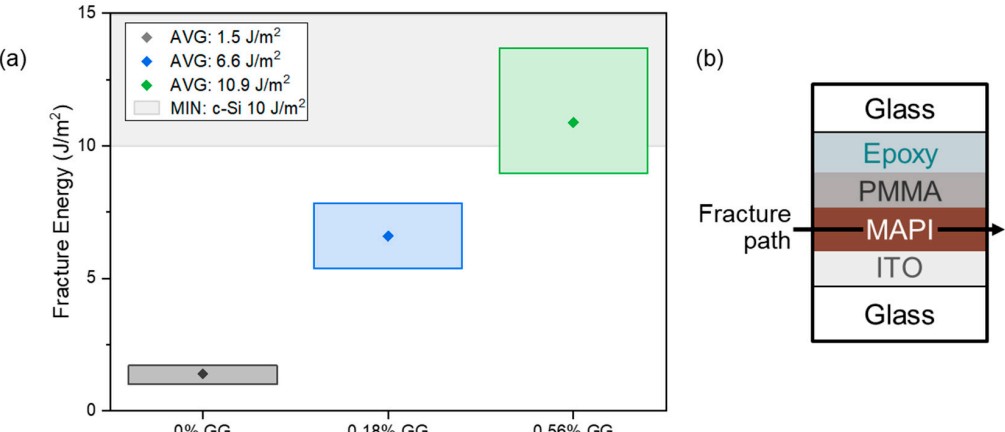

**Figure 6.** (**a**) Fracture energies collected from double cantilever beam testing for spin-coated 0% GG in an inert environment and ambient blade-coated 0.18% and optimized 0.56% GG given with respect to the range of fracture energies for crystalline silicon cells, >10 $J/m^2$ [30]; (**b**) schematic of a double cantilever beam testing sample demonstrating fracture occurring through the PVSK.

## 4. Conclusions

Our findings suggest that the addition of an optimized concentration of GG, 0.56 wt%, allows the fabrication of uniquely stable and durable quench-free perovskite films in ambient conditions. This is enabled through an increase in viscosity while maintaining Newtonian characteristics, thus balancing the crystal nucleation and growth regimes for efficient deposition, upwards of 25 mm/s. This also induces a smooth compact surface with over a 40% reduction in root-mean-square roughness compared to spin-coated control films, as shown through optical images and AFM measurements. These morphological benefits are accompanied by a proper I-V response in 0.56% GG films as validated through lock-in thermography, as well as enhanced optoelectronic and chemical stability under humidity and heat conditions for 12 h at 85% RH or 85 °C validated through XRD and PL measurements, all the while boasting one of the highest fracture energies reported for perovskite materials, making these films scalable, stable, and durable.

With the help of the biopolymer additive and efficient meniscus-driven blade coating, the crystallization process can be effectively altered. Supersaturation rates can be tuned to produce a unique range of morphologies with variable domain sizes and degrees of surface texturing according to additive concentration and without the use of excessive heat. The result is a reduction in detrimental tensile thermal stresses, which can advance degradation in the film. By altering the rheology and free energy considerations of the PVSK ink, the radial, spherulitic growth of similarly oriented domains occurs in a space-filling nature while concentrating defects, impurities, and possibly the additive to domain boundaries. This serves as one possible mechanism of self-passivation behind the improved resilience to heat and humidity aging.

Most notably, this work shows a systematic means of investigating key stability characteristics derived from a tunable morphology for the scalable, quench-free production of mechanically and environmentally robust PVSK films. For the first time, less than one weight percent addition of a polysaccharide biopolymer has been able to convert an ink optimized for use with antisolvent spin coating in an inert environment into a quench-free ink capable of being processed in open-air via a scalable deposition method. A key implication of this work is to hopefully accelerate the commercialization of PVSK solar cells through the usage of a single, common PVSK solvent in ambient conditions without a quenching step. In doing so, we hope to enable a scalable manufacturing process of robust perovskite solar cells by having shown how crystallization theory can be leveraged to improve morphology, stability, and durability.

**Supplementary Materials:** The following supporting information can be downloaded at: https://www.mdpi.com/article/10.3390/en17061455/s1, Figure S1: Illuminated lock-in illumination amplitude and phase images; Figure S2: XRD spectrums after heat aging on an extended 2θ axis; Figure S3: GIWAXS 2D spectrums; Figure S4: Preparation process schematic; Figure S5: Crystallization process flow schematic; Figure S6: Images of blade-coated samples on varying substrate sizes; Table S1: Nomenclature list. Table S2: Series and shunt resistances of blade-coated samples.

**Author Contributions:** Conceptualization, C.C. and N.R.; Funding acquisition, N.R.; Investigation, C.C., S.P., A.G., K.B., M.A., M.L. and R.Z.; Methodology, M.M.; Project administration, N.R.; Supervision, N.R.; Writing—original draft, C.C. and S.P.; Writing—review and editing, S.P., A.G., K.B., M.A., M.L., R.Z., A.R. and N.R. All authors have read and agreed to the published version of the manuscript.

**Funding:** This material is based upon work supported by the U.S. Department of Energy's Office of Energy Efficiency and Renewable Energy (EERE) under Solar Energy Technologies Office (SETO) Agreement Number DE-EE0010502.

**Data Availability Statement:** The original contributions presented in the study are included in the article/Supplementary Material, further inquiries can be directed to the corresponding author. Some data presented in this article may not be readily available because the data are part of an ongoing study.

**Acknowledgments:** The authors acknowledge resources and support from the Advanced Electronics and Photonics Core Facility at Arizona State University. The authors acknowledge the use of facilities within the Eyring Materials Center at Arizona State University supported in part by NNCI-ECCS-1542160.

**Conflicts of Interest:** The authors declare no conflicts of interest. The funders had no role in the design of the study; in the collection, analyses, or interpretation of data; in the writing of the manuscript; or in the decision to publish the results.

## Appendix A. Characterization

Bruker Multimode 8 Atomic Force Microscope was used to image the surface of our samples. The images were recorded at a 5 μm × 5 μm resolution with a scan rate of 1 Hz and 256 lines/sample. The Keyence VHX-7000 Microscope was used to verify the morphology of the sample using the transillumination settings, and a smartphone was used to record macroscale 1 × 1 in images.

Illuminated lock-in thermography (ILIT) uses pulsed light and thermal imaging over time to produce thermal amplitude maps of a sample [31,32]. The amplitude of the thermal emission is directly related to the amount of non-radiative recombination occurring at the point; the thermal emission is influenced by the light absorption and lateral carrier transport within the material [30]. ILIT was measured using a modified ThermoSensorik GmbH ThermoSensor (Erlangen, Germany). The tool's 4-quadrant power supply was connected to a green LED array from Brightspot Automation, which was used to pulse light onto the sample. The image resolution is approximately 5-μm per pixel. Due to uncertainties in the location of the light relative to the sample and in calibration, the absolute amplitude (in mK) is not used in the thermal amplitude maps. Rather, a relative measure of the maximum observed thermal amplitude is used. In this way, we comment on the distribution of thermal emission across the sample, which changes with film morphology, rather than an exact measure of heating at a given point.

The I-V responses of the samples were measured using PAIOS, an all-in-one measurement equipment for photovoltaic devices and LEDs, and photoluminescence (PL) was measured using an in-house BLACK-Comet UV–Vis Spectrometer from StellarNet with a laser wavelength of 425 nm.

The samples were aged in the Thermotron Model SM-8-8200 Environmental Test Chamber (Thermotron, Holland, MI, USA), and the fracture data was collected through the use of a double cantilever beam set up which features the Delaminator Adhesion Test System in the configuration. For fracture testing, sandwich-like structures were assembled with perovskite films coated on ITO glass being covered with a protective layer of polymethyl methacrylate (PMMA) which was then epoxied to an additional glass slide. A manually created pre-crack was formed to assist initial crack formation at the perovskite layer during the uniaxial cyclic loading process, and the average $G_c$ was calculated from averaging multiple critical load values from the resulting load-displacement curve as the crack propagated along the length of the sample.

Grazing-incidence wide-angle X-ray scattering (GIWAXS) measurements were performed on a Xenocs Xeuss 3.0 SAXS/WAXS instrument. A GeniX3D Cu High Flux Very Long (HFVL) focus source was used to produce an 8 KeV Cu K alpha collimated X-ray beam with a wavelength of 1.541891 Å (generated at 50 kV and 0.6 mA). A windowless EIGER2 R 1M DECTRIS Hybrid pixel photon counting detector was used to collect the scattering signal at the sample-to-detector distance of 80 mm to cover a Q range between ~0.5 Å$^{-1}$ and ~3.5 Å$^{-1}$. Default GIWAXS lineup and rectangular beam (0.8 mm × 1.2 mm) were used for measuring each sample. Grazing incidence angles were set as 0.2° and 1°. Measuring time was 30 min for each sample. Q representation images (Qx vs. Qz) were reduced from the obtained images from the 2D detector.

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
