# Peer review of "Scalable and Quench-Free Processing of Metal Halide Perovskites in Ambient Conditions"

_energies, doi:10.3390/en17061455_

Round 1

Reviewer 1 Report

Comments and Suggestions for Authors

This article is devoted to the current issue of developing new materials for perovskite photovoltaics. Its topics correspond to those of the Eneries MDPI journal.

However, in my opinion, there are several significant shortcomings in the work. The main one is the lack of testing of solar cells based on the developed material. In addition, the stability of these films has not been properly studied. The authors mainly study the phase composition by X-ray diffraction and luminescence for this purpose. However, these methods do not provide an idea of the key electrophysical properties of the films that affect the efficiency of the solar cell. The current-voltage curves given by the authors (Fig. 3d) also do not say anything about the change in properties over time. The difference in the course of these curves depending on the composition seems obvious. In addition, as follows from the “Supplemrntary materials” section, the thickness of the films also differed, this should also affect the measured current. It is also worth noting that when studying electrical properties, it would be correct to give specific values. Without all this, it is not clear how applicable the films obtained by the authors are for creating solar cells

Author Response

Thank you for your helpful feedback. Please see the attached document for detailed responses. 

Reviewer 2 Report

Comments and Suggestions for Authors

General comment:

In this manuscript, the authors presented a study of adding gellan gum to control the crystallization of halide perovskite films. This study was interesting. Accordingly, I would like to recommend this article. Selected comments go as follows.

Comment 1:

There is no experimental section. The author should make it up.

Comment 2:

Some words were missing in the caption of Figure 1.

The captions of Figure 2, 4, and 5 are too long. The authors should make them concise.

Comment 3:

No conversion efficiency was mentioned. The author should provide it.

Author Response

(The authors gave the same response as above.)

Reviewer 3 Report

Comments and Suggestions for Authors

Thank you for your efforts in preparing the manuscript. The topic is interesting and promising in the field of solar cell material sciences.The materials and methodology are not in the main part of the manuscript. To ensure meeting the standards of the energies, I suggest the whole article should be revised and properly presented in alignment with current template. Additional proper support and evidences from the literature to be included while discussing the results.

In this manuscript, there are many issues to consider and that can be used for improving the quality of the manuscript, as below:

1- The abstract needs to be written by standard way. Add the impact of the study in the solar energy field, the method, and the results.

2- To easily follow up the idea, a graphical abstract or a graphical representation of the components emphasizes by this study from preparation to fabrication to testing to be included.

3- The introduction section should be expanded to reflect the impact of the study on the energy and energy efficiencies, to reflect the scope of the journal. Also in this part, the authors to indicate how their study contributes to the field in comparison to other existing studies. The author may use the below articles, or others, to support the discussion:

a) Scalable Fabrication of Metal Halide Perovskite Solar Cells and Modules https://pubs.acs.org/doi/10.1021/acsenergylett.9b01396

b) Lead metal halide perovskite solar cells: Fabrication, advancement strategies, alternatives, and future perspectives. https://www.sciencedirect.com/science/article/abs/pii/S235249282300377X?via%3Dihub

 c) https://pubs.acs.org/doi/10.1021/acsenergylett.9b01396

4- The materials and methodology sections are missing from the main body of the manuscript. No clear indication of the selection of tools and materials as the soul of the study– The author should present all those in the body of the manuscript.

5- Many questions raised while reading this manuscript, even very tough to follow, due to the structure used.  Clarify/justify the selection of the concentrations in figure 1. What happens if the concentration is more than 2.29%. The approach of the optimum concentration is not clear and not even discussed and compared with the literature. What is the ultimate target of this study? In figure 3, it was expected to see the 2.29% of GG in figure 3. Clarify why it is not included.

6-Add schematic diagram on the preparation process or the Growth diagram of the material fabrication.

7- Add the abbreviation and nomenclature lists; such as Cs, Cmin, Cmax, C, PVSK, GG, XRD, PL, MAPI, ITO …etc.

8- The results section can be improved. Authors to analyze the work with existing and compare results to the studies that related solar module parameters such PCE, flux, voltage, and FF to the fabrication method. Additionally, did the author assess the solar or energy efficiency of the material produced and effect of the area/size of the samples?

9- Add optical images of the samples produced for comparison purpose.

10- Lines 154-156 remove the italic font. This is valid for the whole manuscript.

11- Line 188 – Figure S1 has been introduced which is in the appendix or supplementary data. I suggest to include in the body or not to include the figure number just highlight the outcomes and refer it to the supplementary information.

12- In figure 4 the colored border of the images corresponds to the time. This should be explained in the text in the results. Additionally, the aging in humidity results in this figure has been introduced before explaining the test (lines 210-215), adjust the discussion accordingly.

13- Lines 212 -214, the authors indicated that they tested the samples in humid air in one place and in another line heating in vacuum. It is confusing, I suggest this to clearly be explained and clarified.

14- The conclusion to be improved. In particular, the authors claimed that they used the optimized concentration of GG but this has not been properly discussed or introduced in the body of the manuscript.

Thank you again for your efforts and good luck.

Author Response

(The authors gave the same response as above.)

Round 2

Reviewer 1 Report

Comments and Suggestions for Authors

In my opinion, the I-V curves in Fig.3d can be described in more detail. For example, the series and shunt resistance was assessed for each of the samples, taking into account their thickness. Otherwise, there are no significant comments

Author Response

We thank the reviewer for their additional feedback and have addressed the comment with an updated manuscript and supplementary table.

Reviewer 3 Report

Comments and Suggestions for Authors

Thank you for your efforts in improving the manuscript.

Author Response

Our pleasure, thank you kindly for reviewing our paper!